**Data Availability Statement:** Data cannot be shared publicly as consent procedures for

# The ARMADILLO text message intervention to improve the sexual and reproductive health knowledge of adolescents in Peru: Results of a randomized controlled trial

Jose E. Perez-Lu[1], Fiorella Guerrero[1], César P. Cárcamo[1], Mónica Alburqueque[1], Marina Chiappe[1], Michelle J. Hindin[2], Ndema Habib[3], Lale Say[3], Lianne Gonsalves[3]*, Angela M. Bayer[1]

**1** Facultad de Salud Pública y Administración, Universidad Peruana Cayetano Heredia, Lima, Peru, **2** Reproductive Health Program, Population Council, New York City, New York, United States of America, **3** Department of Sexual and Reproductive Health and Research Including the UNDP/UNFPA/UNICEF/WHO/World Bank Special Programme of Research, Development and Research Training in Human Reproduction, World Health Organization, Geneva, Switzerland

* gonsalvesl@who.int

## Abstract

### Background

The ARMADILLO Study determined whether adolescents able to access SRH information on-demand via SMS were better able to reject contraception-related myths and misconceptions as compared with adolescents receiving pushed SMS or no intervention.

### Trial design

This trial was an unblinded, three-arm, parallel-group, individual RCT with a 1:1:1 allocation. Trial registration: ISRCTN85156148.

### Methods

This study was conducted in Lima, Peru among participants ages 13–17 years. Eligible participants were randomized into one of three arms: Arm 1: access to ARMADILLO's SMS information on-demand; Arm 2 access to ARMADILLO SMS information pushed to their phone; Arm 3 control (no SMS). The intervention period lasted seven weeks. At baseline, endline, and follow-up (eight weeks following endline), participants were assessed on a variety of contraception-related myths and misconceptions. An index of myths-believed was generated. The primary outcome assessed the subject-specific change in the mean score between baseline and endline. Knowledge retention from endline to follow-up was also assessed, as was a 'content exposure' outcome, which assessed change in participants' knowledge based on relevant SMS received.

### Results

In total, 712 participants were randomized to the three arms: 659 completed an endline assessment and were included in the primary analysis. Arm 2 participants believed fewer

participants and the trial protocol did not include making survey results publicly available. Data are securely stored on the HRP e-Archive system, part of World Health Organization's (WHO) official Enterprise Content Management (ECM) platform and are subject to WHO policy of data use and data sharing. The data used for this analysis can be made available on reasonable request to SIS@who. int, with Subject: 'Inquiry on ARMADILLO Peru study data'.

**Funding:** The ARMADILLO Study was implemented with the financial support of the UNDP-UNFPA-UNICEF-WHO-World Bank Special Programme of Research, Development and Research Training in Human Reproduction (HRP), a cosponsored programme executed by the World Health Organization (WHO). The corresponding author (a member of HRP) was the global coordinator of the study and was involved in the study's design and interpretation of the data and drafting of the manuscript. Data collection and analysis, as well as manuscript drafting was led by the Universidad Peruana Cayetano Heredia team. The corresponding author had full access to all the data in the study and final responsibility for the decision to submit for publication.

**Competing interests:** The authors have declared that no competing interests exist.

myths at endline compared with control arm participants (estimated subject-specific mean difference of -3.69% [-6.17%, -1.21%], p = 0.004). There was no significant difference between participants in Arm 1 vs. the control Arm, or between participants in Arm 1 vs. Arm 2. A further decrease in myths believed between endline and follow-up (knowledge retention) was observed in all arms; however, there was no difference between arms. The content exposure analysis saw significant reductions in myths believed for Arm 1 (estimated subject-specific mean difference of -9.47% [-14.83%, -4.11%], p = .001) and Arm 2 (-5.93% [-8.57%, -3.29%], p < .001) as compared with the control arm; however Arm 1's reduced sample size (n = 28) is a severe limitation.

## Discussion

The ARMADILLO SMS content has a significant (but small) effect on participants' contraception-related knowledge. Standalone, adolescent SRH digital health interventions may affect only modest change. Instead, digital is probably best used a complementary channel to expand the reach of existing validated SRH information and service programs.

## Background

Worldwide, there are almost 1.8 billion adolescents and youth between the ages of 10 and 24, representing nearly one quarter of the global population and in many countries' populations, they are an even higher proportion [1]. In Peru, there are approximately 8.7 million adolescents and youth ages 10–24 who represent 28% of the total population [2]. Adolescents often face challenges to their health and wellbeing, especially related to their sexual and reproductive health (SRH). In Peru, the rate of adolescent pregnancy has remained unchanged over time and continues to be unintended for most adolescents. According to the 2018 Peru Demographic and Health Survey (DHS/ENDES), 12.6% of 15–19 year old females were currently pregnant or parenting [3], a percentage that has not changed significantly since 1991/92 [4]. In 2016, 66% of Peruvian 15–19 year olds who had been pregnant reported that their last pregnancy was unintended, a percentage that has steadily increased over time [5].

Since 2006, the Peruvian Ministry of Health (MOH or MINSA) has expanded targeted health services for adolescents [6]. However, the Peruvian Ombudsman's Office (*Defensoría del Pueblo*) recently found that, in practice, these adolescent-specific health services provide services for very limited hours and require parental accompaniment, running counter to current legislation [7]. There is a need, therefore, for innovative strategies to effectively engage with Peruvian adolescents, providing them with SRH information and services that are tailored and relevant to them.

Mobile phone-based health interventions have become popular, globally, mirroring increases in mobile phone ownership and use by young people [8]. For example, one study of 24 low- and middle-income countries (LMICs) showed that an average of 83% of youth aged 18–29 owned mobile phones, with a range of 53% to 95% across individual countries [9]. Text messaging is a popular form of mobile phone communication among youth in many parts of the world, with recent studies showing that texting is more popular among people aged 18–34 versus those aged 35 and older [10].

There have been some evaluations of mobile phone SRH interventions targeting youth. One systematic review of 28 digital SRH interventions targeting youth aged 10–24 showed that

youth found text messages to be highly acceptable due to the confidentiality of mobile phone communication, informative, and easy to understand and share [11]. However, only 3 of the 28 studies reviewed were in LMICs and none of the rigorous evaluations (such as randomized controlled trials or RCTs) were in LMICs.

For these reasons, we conducted the Adolescent/Youth Reproductive Mobile Access and Delivery Initiative for Love and Life Outcomes (ARMADILLO), a three-armed RCT to test the effect of an on-demand SMS text message intervention with adolescents [12]. The study was conceived by the World Health Organization's Department of Sexual and Reproductive Health and Research and developed in partnership with the Universidad Peruana Cayetano Heredia (UPCH) in Lima, Peru, and technology partner Ona, based in Kenya, where we had a parallel study.

This manuscript focuses on the primary and selected secondary results of ARMADILLO Peru. The primary objective was to determine whether adolescents aged 13–17 able to access SRH information on-demand via SMS are better able to reject contraception-related myths and misconceptions as compared with adolescents receiving pushed SMS or usual care (no intervention). As a secondary objective we assessed participants' retention of contraception-related information. Finally, we explored whether there was a content exposure effect, linking the specific content and quantity of SMS an adolescent participant requested/received to their ability to reject corresponding contraception-related myths and misconceptions.

## Methods

### Study design

This was an unblinded, three-arm, parallel-group, individual RCT with a 1:1:1 allocation. The full procedures for the ARMADILLO trial (registration number: ISRCTN85156148) are described elsewhere [12].

### Participants

The study was conducted in Peru's capital city Lima, in Pampas, one of the seven zones of San Juan de Miraflores, in one of Lima's 43 districts. The inclusion criteria were as follows: adolescents (males or females) ages 13–17; literate; have their own mobile phone (meaning that it is primarily in their possession and that they control when and with whom they share access) and report regular use of the phone; had a mobile phone with them during recruitment; and report current use of text messaging. The study population consisted of randomly selected male and female adolescents who met the eligibility criteria.

### Recruitment and baseline assessment

First, a household census was carried out to enumerate all eligible adolescents in the study zone. For the census: first, all of the blocks in the study zone were numbered and the blocks to be sampled were randomly selected; and next, all of the houses in the selected blocks were numbered and the houses to be sampled were randomly selected. Then, one eligible adolescent from each household was randomly selected for recruitment. If that person was not eligible, or not at home, and there was a second eligible person, that person was selected. Otherwise, the next household was used until the sample was completed. Each household was visited twice to find the selected youth before proceeding to the next randomly selected household. Adolescents who agreed to participate, after gaining parental/guardian consent, completed a baseline survey, generally during the same home visit, in a private place in the home, and in the presence of only the data collector.

## Intervention and control groups

Arms 1 and 2 received the ARMADILLO intervention, which consists of SMS with content related to seven SRH domains or topics, each of which included different sub-domains or sub-topics, as shown in Fig 1. The ARMADILLO domains and sub-domains and the accompanying SMS were identified, developed and validated through a highly participatory process with youth in different regions of Peru, as described elsewhere [13]. Messages in Domain 5, or "How can I protect myself?", centered on contraception; the trial's primary and secondary outcome analyses presented here are the result of participants' engagement with this domain in particular.

In Arm 1 (on-demand), participants received access to one new SRH domain each week. This arm initiated the week with a "domain welcome message" that included a number-based menu that gave participants the ability to request 12–25 SMS with information on several sub-domains related to the week's SRH domain. The week closed with an SMS-based quiz. In Arm 2 (push), participants received access to one new SRH domain each week. This arm initiated the week with a "topic alert message," followed by 10 SMS (2 SMS per day) with information on several sub-domains related to the week's SRH domain, which were automatically pushed to the participant's phone. On Day 6, participants had the option to 'respond to read more' to immediately receive 2–3 additional SMS from this domain. The week closed with an SMS-based quiz. In Arms 1 and 2, each participant initiated the intervention with a different domain, which was selected at random, and then cycled through the remaining domains sequentially. If participants responded to the weekly quiz, they received the equivalent of approximately $1 USD in credit for their mobile phone. In Arm 3 (Control), participants received "routine care," that is, no SMS.

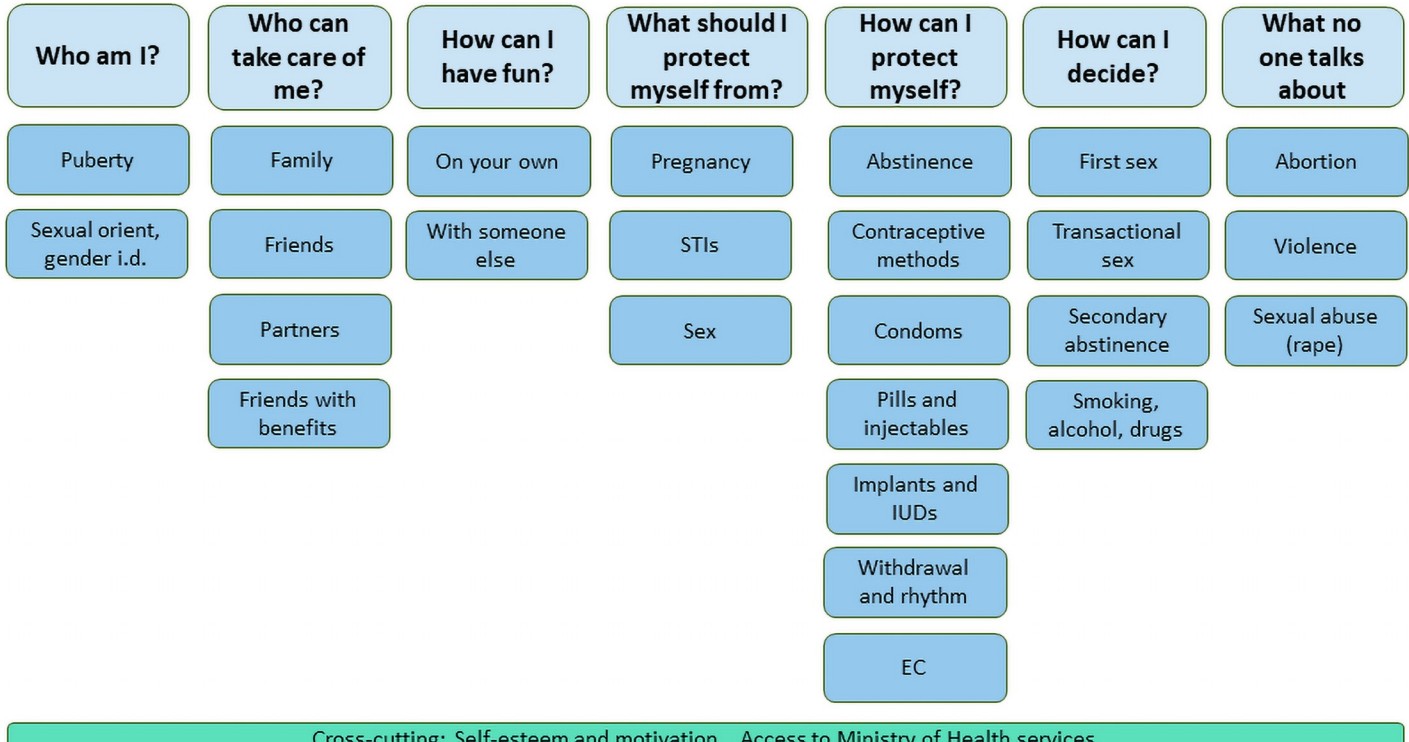

**Fig 1. Domains and sub-domains for the ARMADILLO Peru SMS intervention.**

The ARMADILLO system was developed on RapidPro (an open-source communication platform developed by Nyaruka, Inc. and UNICEF) and was hosted by the technology partner, Ona (Nairobi, Kenya).

## Primary outcome

The primary outcome of the RCT was the subject-specific change in the mean score on the "index of myths and misconceptions around contraception" between baseline and endline (7 weeks following intervention start). The change in myth score was computed for all participants across the three arms. The index contained 17 items that measured common myths/misinformation related to condoms and contraceptive methods (traditional and modern). These included correct use, side effects, appropriateness for adolescents, and effectiveness at preventing pregnancy and HIV and other sexually transmitted infections (STIs). A "percentage incorrect" was created based on the participant's number of incorrect answers to the 17 items corresponding to the percentage of myths and misconceptions believed.

## Secondary and exploratory outcomes

The secondary outcome was knowledge retention from endline to follow-up (8 weeks post-endline). The exploratory dose response outcome linked change in knowledge to the relevant SMS received. Both outcomes used the contraception myths and misconceptions index. The retention outcome was the mean change in the score on the index between participants across the three arms at endline versus follow-up. In the retention outcome, the 100-point score generated for the index was calculated in the same way for all participants, as described above. The content exposure outcome was the mean change in the score on the index between participants across the three arms at baseline versus endline. Here, the 100-point score was created based on each participant's number of incorrect answers to the survey items that corresponded to the SMS messages the participant received. In other words, the survey questions used to generate the score related specifically to the SMS that participants requested/received during the intervention. For example, if the participant received SMSes relating to 10 out of the 17 survey items, only those corresponding survey items were used to create the score.

## Baseline characteristics and outcomes assessments

All participants completed surveys at three time points: 1) baseline, immediately prior to randomization; 2) endline, 7 weeks following the start of the intervention period; and 3) follow-up, 8 weeks following the endline survey. The survey instrument included questions about sociodemographic characteristics; mobile phone use; sexual behavior and condom and contraceptive use; and scales and indices of measures of knowledge and attitudes related to the SRH domains included in the intervention. These scales and indices were identified through extensive literature review, and measures that had been validated with similar populations were used whenever possible.

Surveys were administered using tablets programmed with Open Data Kit. Most of the survey was administered as a face-to-face interview, during which trained survey interviewers asked participants the questions and the interviewers entered the participants' responses into the tablets. At baseline, for sensitive questions about sexual behaviors, the survey interviewer handed the tablet to the participant and responses were entered by the participants themselves.

## Sample size

A sample size of 705 participants was required to provide 80% power to detect a 10% change in the mean number of myths believed from baseline to endline, based on an assumed mean

baseline belief in the myths index of 0.55 (an estimate developed taking into account findings from other studies [14, 15]), with a standard deviation of .30 and a dropout rate of up to 20%. The sample size was calculated to allow for pairwise comparisons between all three arms. Slight increases in sample sizes were to allow for balance in sample size across age groups and for males and females.

## Randomization

For randomization, the research team randomized the three study arms into a sequence numbering 1 to 720. Then, they numbered 720 envelopes and placed a label with the study arm corresponding to that number inside. The envelopes were sealed and the contents were unknown to the data collectors at the point of randomization. Immediately after the baseline survey, an envelope was opened and eligible adolescents were randomized to one of three groups using 1:1:1 allocation. Entry into one of the three arms—marking the start of that participant's intervention period—began automatically the following day, when participants received their first domain welcome menu (Arm 1, on-demand), topic alert (Arm 2, push), or no message (Arm 3, control).

## Data analysis

**Analysis populations.** All outcomes were analyzed as intention-to-treat (ITT), which included all participants randomized into the study who completed baseline and endline surveys, independent of "treatment" (if any) received and/or level of compliance. Additionally, we conducted a per-protocol analysis for the primary and secondary outcomes. This included the randomized population that engaged with the intervention as described in the protocol.

The PP analysis assessed adherence to the intervention. To assess whether the participant was actually engaged in the intervention, for on-demand (Arm 1), a participant must have received the outcome-affiliated Domain 5 "welcome message" *and requested* one or more informational SMS's. For Arm 2, a participant must have received the Domain 5 "topic alert message" (welcome message equivalent) and 10 pushed, informational SMS.

**Statistical analysis.** Participants' baseline characteristics, and the results for the primary and secondary outcomes, were summarized for all randomized participants according to treatment allocation. Categorical variables were summarized using the number and proportion of participants and quantitative normally distributed variables were presented using the mean and standard deviation (SD).

The mean change in the score on the contraception myths and misconceptions index was treated as a continuous variable, with the comparison between study arms carried out using parametric tests if normality assumptions held. A generalized linear model (GLM) using a normal distribution and identity link was used to compare scores between arms while adjusting for the time (days) in which the follow-up survey was conducted. These analyses were carried out for the ITT and PP populations. All tests were two-sided with a significance level of 5%. Stata/IC v12 (Stata Corp., College Station, TX, USA) was used for analysis.

## Ethical considerations

This study was approved by the ethics committees of the World Health Organization (A65892b) and the Universidad Peruana Cayetano Heredia (399-12-17). All adolescent participants provided their written informed consent prior to participating in project activities. Parents and guardians provided their written informed consent for their child's participation.

### Role of the funding source

The ARMADILLO study was funded by the UNDP/UNFPA/UNICEF/WHO/World Bank Special Programme of Research, Development and Research Training in Human Reproduction (HRP). The corresponding author (a member of HRP) was the global coordinator of the study and was involved in the study's design and interpretation of the data and drafting of the manuscript. Data collection and analysis, as well as manuscript drafting was led by the Universidad Peruana Cayetano Heredia team. The corresponding author had full access to all the data in the study and final responsibility for the decision to submit for publication.

## Results

### Study participants

As show in Fig 2, the household census identified 1,798 adolescents aged 13–17 that lived in "Pampas." Of these, 1,086 were excluded because they did not meet the inclusion criteria (845) or declined to participate (241). Of those that did not meet inclusion criteria, 464 (54.9%) were deemed ineligible as they did not possess their own mobile phones. Ultimately, 712 participants were randomized to the three arms. Of these 712, 659 completed an endline assessment and were included in the ITT analysis. Of these 659, 599 completed a follow-up assessment. The PP analysis includes the 469 participants that met the criteria described under "Analysis populations." Of these participants, 428 completed a follow-up assessment.

At baseline, demographic characteristics and history of sexual activity appeared balanced across participants in the three study arms at baseline (Table 1).

**Primary outcome: Change in myths/misconceptions believed from baseline to endline.**   Across arms, participants believed an average of 46.8% of contraception myths and misconceptions at baseline and 43.1% at endline. The distribution of mean index scores across the three arms at baseline and endline appeared similar (Table 2). In the adjusted ITT analysis, participants who received push notifications had a significant decrease in the number of myths believed from baseline to endline as compared with participants in the control group (difference in mean reduction of -3.68%, 95% CI -6.14% to -1.22%, p = 0.003). There was no difference between participants who received information on-demand and the controls, or between the two intervention arms (Table 2).

### Secondary outcome: Knowledge retention

In the adjusted analysis for knowledge retention, which measured change in myths and misconceptions between the endline and follow-up survey, the mean score across all participants was 40.1% at endline. Each arm saw a small, significant reduction in the number of myths and misconceptions believed between endline and follow-up; however there was no significant difference in these reductions between arms (Table 2).

### Exploratory outcome: Content exposure

In this analysis, for which each participant's survey questions were matched to the SMSes the participant received, across arms, participants believed an average of 47.7% of contraception myths and misconceptions at baseline. The criteria for participants to be included in this analysis resulted in a very small sample size for the on-demand arm (n = 28), limiting the ability to interpret results from this arm. There was no difference in the baseline mean index scores across the three arms (Table 2). In the adjusted analysis, there were significant improvements for both intervention arms—compared to controls, on-demand participants saw a 9.47% reduction in the number of myths believed (95% CI -14.83% to -4.11%, p = .001), while push

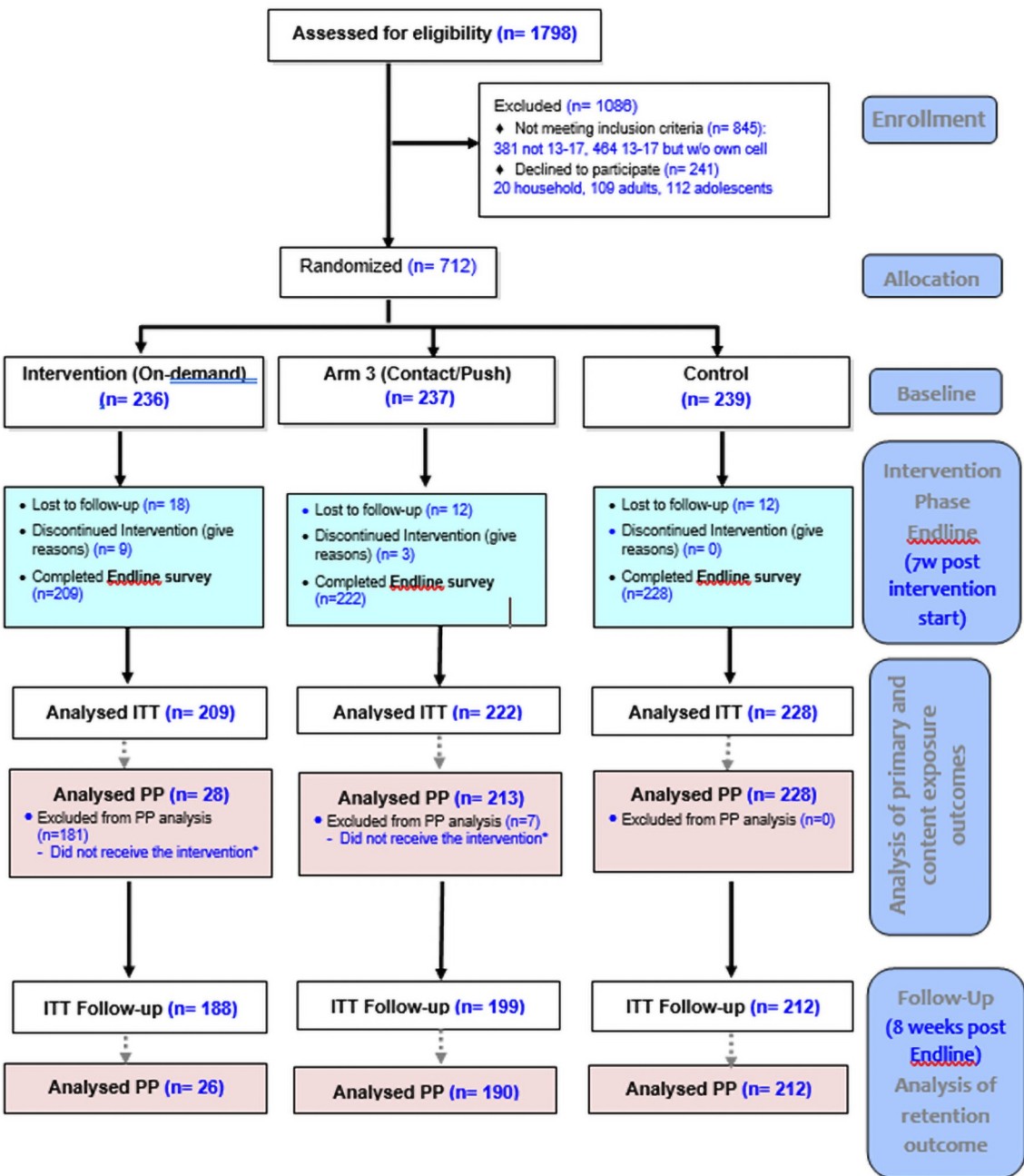

**Fig 2. ARMADILLO Peru CONSORT flow diagram.** *Participant that received the intervention is defined as someone who received text messages contained in ARMADILLO'S Domain 5: 1. For Arm 1 (on-demand): one Domain 5 menu ("welcome message") AND requested 1+ message(s) 2. For Arm 2 (push): one Domain 5 message ("alert message") AND the domain's 10 informational messages successfully pushed.

**Table 1. Baseline characteristics, by randomization arm, of the study population (female and male 13–17 year olds, San Juan de Miraflores, Lima, Peru) (n = 712).**

|  | Arm 1: On-demand | Arm 2: Push | Arm 3: Control | Total |
|---|---|---|---|---|
| Randomized (N) | 236 | 237 | 239 | 712 |
| **Age** |  |  |  |  |
| Mean (SD) | 15.13 (1.38) | 15.41 (1.28) | 15.17 (1.37) | 15.24 (1.35) |
| **Gender** |  |  |  |  |
| Male, n (%) | 109 (46.2) | 120 (50.6) | 104 (43.5) | 333 (46.8) |
| Female, n (%) | 127 (53.8) | 117 (49.4) | 135 (56.5) | 379 (53.2) |
| **Education level** |  |  |  |  |
| Never attended school, n (%) | 0 (0) | 0 (0) | 0 (0) | 0 (0) |
| Primary complete, n (%) | 2 (0.9) | 2 (0.8) | 0 (0) | 4 (0.6) |
| Secondary complete, n (%) | 206 (87.3) | 206 (86.9) | 214 (89.5) | 626 (87.9) |
| At least some post-secondary, n (%) | 28 (11.9) | 29 (12.2) | 25 (10.5) | 82 (11.5) |
| **Number of household members** |  |  |  |  |
| Mean (SD) | 3.33 (1.07) | 3.22 (1.06) | 3.30 (0.98) | 3.29 (1.04) |
| **Number of children** |  |  |  |  |
| 0 children, n (%) | 236 (100.0) | 235 (99.2) | 233 (97.5) | 704 (98.9) |
| 1 child, n (%) | 0 (0) | 2 (0.8) | 6 (2.5) | 8 (1.1) |
| **History of sexual activity** |  |  |  |  |
| Yes, n/N (%) | 68/233 (29.2) | 57/236 (24.2) | 69/237 (28.9) | 194/706 (27.5) |
| **Age at first sexual intercourse** |  |  |  |  |
| 12 years old or earlier, n/N (%) | 8/68 (11.8) | 5/56 (8.9) | 2/69 (2.9) | 15/193 (7.8) |
| 13–14 years old, n/N (%) | 14/68 (20.6) | 12/56 (21.4) | 15/69 (21.7) | 41/193 (21.2) |
| 15–17 years old, n/N (%) | 46/68 (67.6) | 39/56 (69.6) | 52/69 (75.4) | 137/193 (71.0) |
| **Use of contraceptive method at first sex** |  |  |  |  |
| Yes, n/N (%) | 50/68 (73.5) | 40/56 (71.4) | 48/68 (70.6) | 138/192 (71.9) |
| **Use of condom or contraceptive method at last sex** |  |  |  |  |
| Yes, n/N (%) | 53/68 (77.9) | 45/57 (79.0) | 52/67 (77.6) | 150/192 (78.1) |
| **Contraception myths and misconceptions index score at baseline** |  |  |  |  |
| Mean (SD) | 46.6 (12.1) | 46.9 (12.7) | 47.0 (10.6) | 46.8 (11.9) |
| **Content exposure outcome score at baseline** |  |  |  |  |
| Mean (SD) | 51.8 (23.0) | 48.0 (15.1) | 47.0 (10.6) | 47.7 (13.8) |

participants saw a 5.93% reduction in the number of myths believed (95% CI -8.57% to -3.29%, p<0.001). There was no significant difference between the two intervention arms (Table 2).

## Comparison of content exposure and PP analyses

PP results can be found in Table 3. Here too, ability to interpret results from the on-demand arm is limited by the small sample size (N = 28). Similar to the content exposures analysis, in the primary analysis for the PP population, the difference in mean improvement between the intervention and control groups was larger for Arm 1 (on-demand) than for Arm 2 (push), although the on-demand/control comparison was only borderline statistically significant.

## Discussion

The results presented here show that the ARMADILLO SMS content brought about a significant improvement in participants' knowledge related to contraceptive methods. In the primary intention-to-treat (ITT) analysis, the push arm resulted in a small (3.7%) but significant

**Table 2. Analysis of primary and secondary outcomes comparing difference in scores between intervention arms and survey time period among female and male 13–17 year olds, San Juan de Miraflores, Lima, Peru.**

| Outcome | Group Mean Estimates | | | Estimated Subject-Specific Mean Difference | | | |
|---|---|---|---|---|---|---|---|
| | Baseline Group Mean Score (SE) | Endline Group Mean Score (SE) | Follow-up Group Mean Score (SE) | Unadjusted Mean Difference (MeanΔ[1], 95%CI) | p-value | Adjusted[2] Mean Difference (MeanΔ[1], 95%CI) | p-value |
| Contraception myths and misconceptions index score (endline—baseline assessment) N = 659 | | | | | | | |
| Arm 1: On-demand (n = 209) | 46.6 (0.84) | 43.1 (0.87) | | -3.55% (-5.36%, -1.73%) | <0.001 | | |
| Arm 2: Push (n = 222) | 46.9 (0.85) | 41.0 (0.90) | | -5.83% (-7.72%, -3.94%) | <0.001 | | |
| Arm 3: Control (n = 228) | 47.0 (0.71) | 45.0 (0.77) | | -2.15% (-3.73%, -0.57%) | 0.008 | | |
| Mean (Δ Arm 1)—Mean (Δ Arm 3) | | | | -1.40% (-3.79%, 1.00%) | 0.254 | -1.40% (-3.80%, 1.00%) | 0.254 |
| Mean (Δ Arm 2)—Mean (Δ Arm 3) | | | | -3.68% (-6.14%, -1.22%) | 0.003 | -3.68% (-6.14%, -1.22%) | 0.003 |
| Mean (Δ Arm 1)—Mean (Δ Arm 2) | | | | 2.28% (-0.34%, 4.91%) | 0.088 | 2.30% (-0.33%, 4.93%) | 0.087 |
| Retention of knowledge outcome (follow-up—endline assessment) N = 599 | | | | | | | |
| Arm 1: On-demand (n = 188) | | 42.9 (0.92) | 40.2 (0.94) | -2.63% (-4.51%, -0.74%) | 0.006 | | |
| Arm 2: Push (n = 199) | | 41.3 (0.94) | 37.6 (0.89) | -3.72% (-5.64%, -1.81%) | <0.001 | | |
| Arm 3: Control (n = 212) | | 45.2 (0.80) | 42.4 (0.79) | -2.77% (-4.43%, -1.12%) | 0.001 | | |
| Mean (Δ Arm 1)—Mean (Δ Arm 3) | | | | 0.15% (-2.35%, 2.64%) | 0.908 | 0.13% (-2.37%, 2.64%) | 0.916 |
| Mean (Δ Arm 2)—Mean (Δ Arm 3) | | | | -0.95% (-3.47%, 1.57%) | 0.460 | -0.89% (-3.41%, 1.64%) | 0.492 |
| Mean (Δ Arm 1)—Mean (Δ Arm 2) | | | | 1.10% (-1.59%, 3.79%) | 0.424 | 1.09% (-1.60%, 3.79%) | 0.426 |
| Content exposure outcome (endline—baseline assessment) N = 478 | | | | | | | |
| Arm 1: On-demand (n = 28) | 51.8 (4.35) | 40.2 (4.90) | | -11.58% (-19.79%, -3.36%) | 0.006 | | |
| Arm 2: Push (n = 222) | 48.0 (1.02) | 39.9 (1.09) | | -8.08% (-10.20%, -5.96%) | <0.001 | | |
| Arm 3: Control (n = 228) | 47.0 (0.71) | 45.0 (0.77) | | -2.15% (-3.73%, -0.57%) | 0.008 | | |
| Mean (Δ Arm 1)—Mean (Δ Arm 3) | | | | -9.43% (-14.76%, -4.09%) | 0.001 | -9.47% (-14.83%, -4.11%) | 0.001 |
| Mean (Δ Arm 2)—Mean (Δ Arm 3) | | | | -5.93% (-8.57%, -3.29%) | <0.001 | -5.93% (-8.57%, -3.29%) | <0.001 |
| Mean (Δ Arm 1)—Mean (Δ Arm 2) | | | | -3.50% (-10.14%, 3.14%) | 0.302 | -3.54% (-10.23%, -3.14%) | 0.299 |

Notes:

[1]Δ refers to the subject-specific change in the outcome from timepoint 1 to timepoint 2. 95%CI refers to the 95% confidence interval.

[2]Adjusted estimates control for the time in which the endline and follow-up surveys were conducted. A generalized linear model (GLM) using a normal distribution and identity link was used to compare scores. There was a delay in the timing of the endline survey for many participants, which ranged from 0 to 72 days (mean = 13.30, SD = 11.28). There was also delay in the timing of the follow-up survey for many participants, which ranged from 13 to 121 days (mean = 64.10, SD = 8.97).

improvement in participant knowledge when compared to the control arm. The results of the targeted ITT content exposure analysis showed a stronger effect for the on-demand arm (9.5% improvement) than for the push arm (5.9% improvement), both compared to the control arm.

This suggests that the effect of the intervention seems to be larger when young people are interested in learning about the topic, as demonstrated by greater improvement in the on-demand (where participants could choose the content most interesting to them) versus push

**Table 3. Analysis of primary and secondary outcomes for per-protocol (PP) analysis.** Female and male 13–17 year olds, San Juan de Miraflores, Lima, Peru.

| Outcome | Group Mean Estimates | | | Estimated Subject-Specific Mean Difference | | | |
|---|---|---|---|---|---|---|---|
| | Baseline Group Mean Score (SE) | Endline Group Mean Score (SE) | Follow-up Group Mean Score (SE) | Unadjusted estimates (MeanΔ[1], 95%CI) | p-value | Adjusted[2] estimates (MeanΔ[1], 95%CI) | p-value |
| Contraception myths and misconceptions index score (endline—baseline assessment) (N = 469) | | | | | | | |
| Arm 1: On-demand (n = 28) | 48.1 (2.29) | 41.4 (2.39) | | -6.72% (-11.74%, -1.70%) | 0.009 | | |
| Arm 2: Push (n = 213) | 46.6 (0.87) | 40.8 (0.91) | | -5.83% (-7.75%, -3.90%) | <0.001 | | |
| Arm 3: Control (n = 228) | 47.0 (0.71) | 45.0 (0.77) | | -2.15% (-3.73%, -0.57%) | 0.008 | | |
| Mean (Δ On-demand)—Mean (Δ Control) | | | | -4.57% (-9.40%, 0.26%) | 0.064 | -4.57% (-9.43%, 0.29%) | 0.066 |
| Mean (Δ Push)—Mean (Δ Control) | | | | -3.68% (-6.15%, -1.20%) | 0.004 | -3.69% (-6.17%, -1.21%) | 0.004 |
| Mean (Δ On-demand)—Mean (Δ Push) | | | | -0.90% (-6.51%, 4.72%) | 0.754 | -0.68% (-6.33%, 4.98%) | 0.814 |
| Retention outcome (follow-up—endline assessment) (N = 428) | | | | | | | |
| Arm 1: On-demand (n = 26) | | 41.0 (2.54) | 38.2 (2.38) | -2.71% (-7.20%, 1.77%) | 0.236 | | |
| Arm 2: Push (n = 190) | | 41.0 (0.96) | 37.2 (0.91) | -3.78% (-5.74%, -1.81%) | <0.001 | | |
| Arm 3: Control (n = 212) | | 45.2 (0.80) | 42.4 (0.79) | -2.77% (-4.43%, -1.12%) | 0.001 | | |
| Mean (Δ On-demand)—Mean (Δ Control) | | | | 0.06% (-4.92%, 5.04%) | 0.981 | 0.01% (-4.99%, 5.00%) | 0.999 |
| Mean (Δ Push)—Mean (Δ Control) | | | | -1.00% (-3.55%, 1.55%) | 0.441 | -0.92% (-3.48%, 1.63%) | 0.479 |
| Mean (Δ On-demand)—Mean (Δ Push) | | | | 1.06% (-4.51%, 6.63%) | 0.709 | 0.71% (-4.86%, 6.29%) | 0.802 |

Notes:

[1]Δ refers to the subject-specific change in the outcome from timepoint 1 to timepoint 2. 95%CI refers to the 95% confidence interval.

[2]Adjusted estimates adjust for the time in which the endline and follow-up surveys were conducted. A generalized linear model (GLM) using a normal distribution and identity link was used to compare scores

groups for the content exposure outcome, as well as a trend toward greater improvement in the on-demand versus push groups in the PP analysis. However, any interpretation of on-demand arm results are severely limited by the fact that while most on-demand arm participants received the menu messages, few (28, or 13.4% of participants in the arm) received an additional information message. Finally, there was no significant difference in knowledge retention following the intervention in comparisons across arms. Therefore, the small but significant continued improvement observed in both on-demand and push arms from endline to follow-up cannot be attributed to either intervention.

This study has several limitations. A noticeable percentage of enumerated 13–17 year-olds were ineligible to participate as they did not own their own mobile phones. This may limit the generalizability of our findings. A second limitation is that this study was individually-randomized in a high-density area of housing. Participants in different arms may have lived in closed proximity to each other and contamination across arms (with participants sharing messages) is possible. A final challenge relates to the on-demand participants' interactions with the

intervention. Retrospective review of the system's data indicated that while most on-demand arm participants received the menu messages, few received an additional information message. This results in an important limitation related to the size of the Arm 1 group for per protocol and content exposure analyses—these findings (as well as their generalizability) must be interpreted with caution due to small sample size.

The engagement rate was discovered after the intervention period ended, making it difficult to know with certainty the cause(s). However, as previously documented by the ARMADILLO study's Kenya site, reasons could include: being unsure as to how to engage, being unwilling to engage, or forgetting to engage [16]. Additionally, as this instance of RapidPro was run from an application on an ARMADILLO Study-owned mobile phone, incoming messages from participants that coincided with short- or long-term disruptions in internet/data signal to the study's phone may also have resulted in participants' requests for information being lost.

As to why the low engagement was discovered when it was too late to intervene, a post-mortem analysis revealed a lack of clarity between study partners around whether there was a need to monitor the backend of the digital system during implementation, and whose role that would be. This lack of clarity was likely exacerbated by key partners being separated by language and time zones, complicating easy communication and rapid response. Recent guidance on youth-centred digital health intervention development highlights the importance of identifying appropriate core team members for each stage of an intervention's design and implementation, with defined roles and clear communication as to what tasks each team member is trained to do [17]. In our case, while it was clear who was responsible for developing the digital health intervention (the technology partner), and analyzing data generated by the system (the research partner), there was a lack of clarity on who monitored the system, and what 'monitoring' a digital health intervention entailed: e.g. keeping credit topped up vs phones charged vs tracking a dashboard displaying participants progressing through flows in real time.

This process-related finding that many participants did NOT receive the intervention intended, also speaks to the importance of looking beyond ITT results when analyzing data from digital health RCTs. Agarwal et al (2016) developed guidelines for reporting on health interventions using mobile phones, entitled the 'mobile health (mHealth) evidence reporting and assessment (mERA) checklist' [18]. Among the 16 criteria is 'fidelity of the intervention,' intended 'to monitor system stability, ensure delivery (and possibly receipt) of messages, or measure levels of participant or end-user engagement with the system.' Given the array of technology complications which can (and did) arise, in addition to usual research challenges implementing a multi-week intervention with young adolescents in a community setting, our incorporation of system-generated data allowed us to go beyond ITT analysis to isolate 'real world' effects among participants who received/engaged with the intervention as intended. Our content exposure analysis provided an even more precise assessment of the effect of receiving an SMS on a specific subject on the ability to influence knowledge on that same subject. An additional strength of our study is the inclusion of some young adolescents (defined as those age 14 and under), who are often overlooked in SRH-related research.

Across push and on-demand participants, overall change from baseline to endline was relatively modest—single digit improvements on an index of a variety of contraception myths and misconceptions. For the limited on-demand participants who successfully engaged with the intervention, there may be contextual explanations. On-demand messaging itself is not a popular strategy in Peru. Push messaging services, by contrast, are widespread. Mobile phone owners can be pushed messages from the Ministry of Health and other health partners, and individuals can subscribe for a variety of paid and unpaid services. Our young, on-demand participants may not have intuitively known how to interact with an on-demand system.

What, then, explains the modest baseline-endline change observed in the push arm, successfully implemented in a country where similar services are commonplace? A possible explanation is that digital health hype has outpaced the evidence for adolescent SRH (ASRH) interventions. A 2016 systematic review of ASRH mobile phone interventions identified several studies that purported to improve SRH-related knowledge and behavior [11]. However, in studies that included a control group, a 12-week interactive SMS sex risk reduction intervention for adolescent females showed no significant difference in the change in sex risk outcomes between intervention and control groups [19]. Additionally, a pilot of a 12-week SMS intervention on HIV prevention for at-risk African American young men showed higher sexual health awareness between intervention versus control groups, but no differences in protected sex [20]. Both studies were in a high-income country (the United States) and had small sample sizes (60 or fewer participants). One RCT of a year-long SMS program to promote sexual health among young Australians found significantly improved knowledge about STIs in the intervention versus control arm participants (though the effect size was small) [21]. There was no significant difference in condom use.

An updated systematic review, conducted in 2017, looked across 13 RCTs from mostly high-income countries that provided adolescents with interventions could result in only very limited changes in specific health behavior (e.g. oral contraception adherence) or knowledge [22], and almost all of the evidence included was rated as having low certainty. Finally, the ARMADILLO study was also carried out in Kenya among young people aged 18–24. Here, the change observed in the intervention arms from baseline to endline was not significantly different from that observed in the control arm, meaning that any change could not be attributed to the intervention.

It is becoming increasingly apparent that a digital health intervention *on its own* (be it push or on-demand) may at best affect only modest change in adolescent SRH-related outcomes. Remembering that adolescents' sexual and reproductive health and wellbeing is affected by a variety of ecological determinants [23] provides an explanation for this. These determinants range from laws/policies enabling access to needed services, to community norms as to what SRH-related services or behavior are appropriate for this age group, to individual knowledge and self-efficacy. Additional, rigorously-designed studies are needed to identify exactly where digital health interventions can be most effective in the complex pathway that spans adolescents' SRH knowledge through to behavior change.

That said, a knowledge-targeting digital health intervention alone is likely *not sufficient* to overcome misinformation and norms-setting from friends, family, and communities. Instead, these digital health interventions, in Peru and more broadly, might be better viewed as complementary interventions to be expand the reach of/be implemented alongside of existing in-school and out-of-school comprehensive sexuality education, broader ASRH interventions [24], and sustained interventions to improve community attitudes towards ASRH [25].

## Supporting information

**S1 File.**
(DOCX)

**S1 Checklist. CONSORT 2010 checklist of information to include when reporting a randomised trial**\*.
(DOC)

## Acknowledgments

The manuscript represents the views of the named authors only.

## Author Contributions

**Conceptualization:** Michelle J. Hindin, Lale Say, Lianne Gonsalves.

**Data curation:** César P. Cárcamo, Angela M. Bayer.

**Formal analysis:** Jose E. Perez-Lu, César P. Cárcamo, Ndema Habib, Lianne Gonsalves, Angela M. Bayer.

**Funding acquisition:** Lale Say, Lianne Gonsalves.

**Investigation:** Jose E. Perez-Lu, Fiorella Guerrero, Mónica Alburqueque, Marina Chiappe, Michelle J. Hindin, Angela M. Bayer.

**Methodology:** Jose E. Perez-Lu, Lianne Gonsalves, Angela M. Bayer.

**Project administration:** Jose E. Perez-Lu, Fiorella Guerrero, Mónica Alburqueque, Angela M. Bayer.

**Supervision:** Lale Say, Lianne Gonsalves, Angela M. Bayer.

**Validation:** Angela M. Bayer.

**Writing – original draft:** Jose E. Perez-Lu, Fiorella Guerrero, Lianne Gonsalves, Angela M. Bayer.

**Writing – review & editing:** Jose E. Perez-Lu, Fiorella Guerrero, Mónica Alburqueque, Marina Chiappe, Michelle J. Hindin, Ndema Habib, Lale Say, Lianne Gonsalves, Angela M. Bayer.

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
