## [Decision Letter · Decision Letter 0]

11 May 2021

PONE-D-20-34599

The ARMADILLO Text Message Intervention to Improve the Sexual and Reproductive Health Knowledge of Adolescents in Peru: Results of a Randomized Controlled Trial

PLOS ONE

Dear Dr. Gonsalves,

Thank you for submitting your manuscript to PLOS ONE. After careful consideration, we feel that it has merit but does not fully meet PLOS ONE’s publication criteria as it currently stands. Therefore, we invite you to submit a revised version of the manuscript that addresses the points raised during the review process.

While the Reviewers found the presentation of the results evaluating the ARMADILLO intervention potentially impactful, they also identified some weaknesses, generally, around the presentation of the results and in putting the study into the proper, larger context (e.g., generalizability). Therefore, considering the manuscript further will require a major revision, addressing each of the Reviewers comments.

We look forward to receiving your revised manuscript.

Kind regards,

Scarlett L. Bellamy, ScD

Academic Editor

PLOS ONE

Additional Editor Comments:

After careful review, although both Reviewers identified potential strengths of the highlighted intervention and its potential impact on adolescent health, they also identified a number of weaknesses. Taken collectively, in order to consider this publication further, it will need a major revision addressing each of the reviewers noted general critiques regarding the presentation of the results and in presenting the implications of the findings addressing the effectiveness of the intervention on the sexual and reproductive health knowledge of Peruvian adolescents.

Journal Requirements:

Reviewers' comments:

Reviewer's Responses to Questions

**Comments to the Author**

1. Is the manuscript technically sound, and do the data support the conclusions?

Reviewer #1: Yes

Reviewer #2: Partly

2. Has the statistical analysis been performed appropriately and rigorously? 

Reviewer #1: No

Reviewer #2: Yes

3. Have the authors made all data underlying the findings in their manuscript fully available?

Reviewer #1: No

Reviewer #2: Yes

4. Is the manuscript presented in an intelligible fashion and written in standard English?

Reviewer #1: Yes

Reviewer #2: Yes

5. Review Comments to the Author

Reviewer #1: Abstract

- it is more typical to include a brief background of the study in addition to the study objective (rather than the objective alone) in the 'background' section of the abstract.

Background

- in the first paragraph you probably mean "In Peru, the rates of adolescent pregnancy has remained unchanged over time..."

Methods

- more information should be provided about how the adolescents were selected; it is not sufficiently clear to say "randomly selected male and female adolescents who met the eligibility criteria".

- please check the parameters cited in your sample size calculation. A quick check (e.g. using the calculator at https://www.sealedenvelope.com/power/binary-superiority/) shows that 373 participants per arm i.e. 746 participants in the main treatment and control arm combined, not counting the third arm, are required to demonstrate a change in outcome from 55% to 65% (10% absolute change) with 80% power at the 5% level of significance; and this is before inflating the sample for expected dropouts. If a different approach has been used for this calculation, e.g. one based on the mean number of correct responses, this should be described more clearly.

- additionally, there needs to be some justification for the assumption of a baseline outcome of 55% and the 10% improvement, e.g. previous work suggesting this baseline level of the outcome; important effect size, e.t.c.

- an analysis of a change-from-baseline score should normally include adjustment for the baseline score as a covariate in the regression model (see the 'change from baseline analysis section in: Committee for Proprietary Medicinal Products. Points to consider on adjustment for baseline covariates. Stat Med 2004;23(5):701–9. Available from: http://www.ncbi.nlm.nih.gov/pubmed/14981670). This is equivalent to modelling the score at endline as the outcome adjusting for baseline score (among other covariates if desired). When you make this change in your analysis please update the abstract to indicate that the outcome was the baseline-adjusted change in the mean score. If the data are in 'long' form (with one variable for the knowledge score and a time indicator for the survey), then a model for the score variable as the outcome with the time variable, group variable (i.e. intervention v. control) and a time-group interaction can be used to estimate baseline-adjusted effects (see doi:10.1016/S0140-6736(18)31782-3 for an example of this and the appendix of doi:10.1186/1471-2431-11-109 for a detailed explanation of this approach, ignoring the adjustments for clustering which are not present in ARMADILLO).

Results

- please include 95% confidence intervals when you report the effect estimates in the main text, i.e. estimate, 95%CI and p-value. Do not report SDs for outcomes in the text of the results or in table 2, as these are used for descriptive purposes and not for inference. However, you may (indeed, should) report baseline scores in Table 1 with their SDs.

- Table 2 and Table 3 can and should be improved. They should have the mean scores at endline (with SEs) of each of the three arms in separate columns, and the pairwise unadjusted and adjusted differences with 95%CI and p-values.

Reviewer #2: Thank you for the opportunity to review this manuscript, which describes the results of a rigorously designed study of a SMS to improve sexual and reproductive health knowledge among teens in Lima, Peru. Based on the abstract and the first 14 pages of the submission, I could find little wrong with the study or the paper (though I'd preferred the paper be written in the active voice so that it was clearer who was doing the research rather than it being done by, assumedly, all of the authors at once.)

But on page 14 the authors state as a limitation that of the 236 people randomized into the first of the RCT's three groups "relatively few (28, or 13.4% of participants in the arm)" actually received the full intervention. I'd say that the use of the word "relatively" is a major understatement! Any researcher who claims never to have made a massive blunder in their research at some point is likely a liar, and I think too many of these mistakes never make it to manuscripts, usually because the authors are embarrassed and don't try. So, I appreciate the authors' honesty in including the paragraph about this major mishap on such a big project. But it should not have been relegated to a paragraph about limitations, and this huge problem with the data collection should certainly not have been left out of the abstract, which makes claims about the results of the study and Arm 1 that are extremely suspect with a sample size so small. After the fact, I noticed that the n for the whole study was in the abstract but not the n of each arm.

In addition, I don't think it's a great idea to claim that one of the study's strengths is the use of per protocol analyses that revealed the Arm 1 intervention failure. It's not a strength that no one notice the error until the data analysis. The researchers in Lima should have been monitoring the data flow in real time, and a nearly 90% difference in participation in one arm would have alerted the researchers that something was wrong and needed to be dealt with immediately. Or maybe something else happened -- but whatever it is, it needs to be explained in more detail than what is given.

All this said, I think this paper has potential value simply for the analysis of Arm 2 and Arm 3. But any revision needs to state bluntly and early what the sample size in the arms were and why the discrepancy between Arm 1 and Arms 2 and 3 exists. And since the results of the RCT (with or without Arm 1) are rather unexciting, an examination of what went wrong and the lessons learned would be much more valuable than a formulaic description of a very small, if statistically significant, effect.

6. PLOS authors have the option to publish the peer review history of their article (what does this mean?). If published, this will include your full peer review and any attached files.

Reviewer #1: No

Reviewer #2: No

---

## [Author Response · Author response to Decision Letter 0]

9 Jul 2021

These are included as a separate word document, uploaded (as requested in the Decision Letter) along with tracked changes and clean copies of the revised protocol.

Best regards,

Lianne on behalf of the authors.

---

## [Decision Letter · Decision Letter 1]

20 Oct 2021

PONE-D-20-34599R1The ARMADILLO Text Message Intervention to Improve the Sexual and Reproductive Health Knowledge of Adolescents in Peru: Results of a Randomized Controlled TrialPLOS ONE

Dear Dr. Gonsalves,

Thank you for submitting your manuscript to PLOS ONE. After careful consideration, we feel that it has merit but does not fully meet PLOS ONE’s publication criteria as it currently stands. Therefore, we invite you to submit a revised version of the manuscript that addresses the points raised during the review process.

As noted in the reviewers comments, you have been very responsive to prior reviewer comments, resulting in a stronger, clearer manuscript. However, there are a few minor lingering issues noted by reviewers requiring further attention. Specifically, Reviewer 1 has some additional recommendation regarding the presentation of summary statistics in Tables and Reviewer 2 has noted where further clarification regarding 'inclusion criteria' should be considered.

We look forward to receiving your revised manuscript.

Kind regards,

Scarlett L. Bellamy, ScD

Academic Editor

PLOS ONE

Journal Requirements:

Additional Editor Comments (if provided):

As noted in the reviewers comments, you have been very responsive to prior reviewer comments, resulting in a stronger, clearer manuscript. However, there are a few minor lingering issues noted by reviewers requiring further attention. Specifically, Reviewer 1 has some additional recommendation regarding the presentation of summary statistics in Tables and Reviewer 2 has noted where further clarification regarding 'inclusion criteria' should be considered.

Reviewers' comments:

Reviewer's Responses to Questions

**Comments to the Author**

1. If the authors have adequately addressed your comments raised in a previous round of review and you feel that this manuscript is now acceptable for publication, you may indicate that here to bypass the “Comments to the Author” section, enter your conflict of interest statement in the “Confidential to Editor” section, and submit your "Accept" recommendation.

Reviewer #1: (No Response)

Reviewer #2: All comments have been addressed

2. Is the manuscript technically sound, and do the data support the conclusions?

Reviewer #1: Yes

Reviewer #2: Yes

3. Has the statistical analysis been performed appropriately and rigorously? 

Reviewer #1: Yes

Reviewer #2: Yes

4. Have the authors made all data underlying the findings in their manuscript fully available?

Reviewer #1: No

Reviewer #2: Yes

5. Is the manuscript presented in an intelligible fashion and written in standard English?

Reviewer #1: Yes

Reviewer #2: Yes

6. Review Comments to the Author

Reviewer #1: If the last two rows of Table 1 are showing mean and standard deviation (SD) like the age row at the top, please indicate this as you have done for other continuous variables in the table.

As indicated in the previous round of comments, Table 1 (descriptive results) should normally report means and SDs for continuous variables - you have done this. However inferential results such as have been reported in Table 2 and 3 should report means and standard errors (SE) not SDs for continuous outcomes. The rest of Table 2 and 3 are fine.

I am satisfied with the other changes made in response to previous reviews.

Reviewer #2: Thank you for the chance to review this revision. The authors did a fantastic job responding to the comments, and I greatly appreciated the detail in the discussion of what went wrong and why. I think this makes the paper more valuable than the data alone. My one suggestion is minor. On page 13 of the revision, the authors state, "The criteria for

inclusion in this analysis resulted in a very small sample size for on-demand arm..." and I eventually figured out what they meant, because of the wording I initially thought they were referring to the inclusion criteria for participants (which was discussed just on the previous page, 12) rather than the criteria for inclusion in the ITT data (explained on page 10). I suggest rephrasing the sentence to be clearer on that issue. Otherwise, this paper looks great. Thank you again for the chance to review the original and revision.

7. PLOS authors have the option to publish the peer review history of their article (what does this mean?). If published, this will include your full peer review and any attached files.

Reviewer #1: No

Reviewer #2: No

---

## [Author Response · Author response to Decision Letter 1]

3 Nov 2021

A document responding to reviewer comments has been uploaded along with the revised manuscript (clean and tracked changes versions)

---

## [Decision Letter · Decision Letter 2]

11 Jan 2022

The ARMADILLO Text Message Intervention to Improve the Sexual and Reproductive Health Knowledge of Adolescents in Peru: Results of a Randomized Controlled Trial

PONE-D-20-34599R2

Dear Dr. Lianne Gonsalves,

We’re pleased to inform you that your manuscript has been judged scientifically suitable for publication and will be formally accepted for publication once it meets all outstanding technical requirements.

Kind regards,

Eugene Demidenko, Ph.D.

Academic Editor

PLOS ONE

Additional Editor Comments (optional):

I read the paper myself and I believe that the paper is in a publishable form. I have just one editorial remark on the citation syntax. (a) Instead of ".[1]" use "[1]." That is, put the period after the citation bracket. (b) Instead of "interventions[24]" use "interventions [24]". That is use the space before the citation bracket.

Reviewers' comments:

Reviewer's Responses to Questions

**Comments to the Author**

1. If the authors have adequately addressed your comments raised in a previous round of review and you feel that this manuscript is now acceptable for publication, you may indicate that here to bypass the “Comments to the Author” section, enter your conflict of interest statement in the “Confidential to Editor” section, and submit your "Accept" recommendation.

Reviewer #1: All comments have been addressed

Reviewer #2: All comments have been addressed

2. Is the manuscript technically sound, and do the data support the conclusions?

Reviewer #1: (No Response)

Reviewer #2: Yes

3. Has the statistical analysis been performed appropriately and rigorously? 

Reviewer #1: (No Response)

Reviewer #2: Yes

4. Have the authors made all data underlying the findings in their manuscript fully available?

Reviewer #1: (No Response)

Reviewer #2: Yes

5. Is the manuscript presented in an intelligible fashion and written in standard English?

Reviewer #1: (No Response)

Reviewer #2: Yes

6. Review Comments to the Author

Reviewer #1: (No Response)

Reviewer #2: All of my questions and revisions have been addressed. Thank you for the ioppurtunity to review your work.

7. PLOS authors have the option to publish the peer review history of their article (what does this mean?). If published, this will include your full peer review and any attached files.

Reviewer #1: No

Reviewer #2: No

---

## [Editor Report · Acceptance letter]

3 Feb 2022

PONE-D-20-34599R2 

The ARMADILLO text message intervention to improve the sexual and reproductive health knowledge of adolescents in Peru: Results of a randomized controlled trial 

Dear Dr. Gonsalves:

I'm pleased to inform you that your manuscript has been deemed suitable for publication in PLOS ONE. Congratulations! Your manuscript is now with our production department. 

Kind regards, 

on behalf of

Dr. Eugene Demidenko 

Academic Editor

PLOS ONE